# The Operative Time for Unilateral Inguinal Hernia Repair in Children Performed with Percutaneous Internal Ring Suturing (PIRS) or Open Approach Method

**DOI:** 10.3390/jcm10061293

**Published:** 2021-03-21

**Authors:** Przemyslaw Karol Wolak, Agnieszka Strzelecka, Aneta Piotrowska, Katarzyna Dąbrowska, Piotr Przemysław Wolak, Ilona Piotrowska, Grażyna Nowak-Starz

**Affiliations:** 1Collegium Medicum, Jan Kochanowski University, ul. Żeromskiego 5, 25-369 Kielce, Poland; strzel@ujk.edu.pl (A.S.); aesculapa@gmail.com (A.P.); ilona.piotrowska@poczta.onet.pl (I.P.); grazyna.nowak-starz@ujk.edu.pl (G.N.-S.); 2Department of Pediatric Surgery, Urology and Traumatology Provincial Hospital, ul. Grunwaldzka 45, 25-736 Kielce, Poland; 3Department of Neonatology and Neonatal Intensive Care, Polish Mothers Health Research Institute, ul. Rzgowska 281/289, 93-338 Łódź, Poland; dabrowskak@yahoo.com; 4City Hospital of Zabrze, ul. Zamkowa 4, 41-803 Zabrze, Poland; zalasna@gmail.com

**Keywords:** inguinal hernia, pediatric, laparoscopic hernia repair, open hernia repair, PIRS

## Abstract

In this study, we compared the operative time for unilateral inguinal hernia repair in children performed with either an open approach (OA) or the Percutaneous Internal Ring Suturing (PIRS) method. It was a retrospective chart review of all patients ages 0 to 18 who underwent unilateral inguinal hernia repair in the Department of Pediatric Surgery, Urology and Traumatology of the Regional Hospital in Kielce between January 2011 and December 2018. Patients with bilateral hernias or additional problems were excluded. Of 878 patients qualified for the study, 701 were in the OA group and 177 in the PIRS group. Overall, the time needed to complete the procedure was significantly longer for the OA method. The operative time was longer if the hernia was left-sided (*p* = 0.024). Analysis by gender showed that operative time was generally longer in males. For both genders, surgery was shorter if the PIRS method was used. For males in the PIRS group the operative time was affected by the location of the hernia, and it was longer for a left-sided hernia. The take-home message is that the PIRS procedure is faster than the OA for inguinal hernia repair in children and it might be considered as a preferred method, especially in females.

## 1. Introduction

A congenital inguinal hernia in the pediatric population is one of the most common conditions requiring surgical treatment, with the reported incidence ranging between 1% and 5% [1,2,3]. It manifests clinically more commonly in younger children, especially premature infants, and the incidence rate is up to 10 times higher in boys. A unilateral inguinal hernia is more prevalent than a bilateral one. Bilaterally patent processus vaginalis is observed in 20% to 40% of patients, and it might be asymptomatic, or it can become clinically significant [4,5,6,7].

Inguinal hernia repair in children can be performed using the open or minimally invasive, laparoscopic approach. One of the methods for laparoscopic inguinal hernia repair is the Percutaneous Internal Ring Suturing (PIRS) procedure described by Patkowski in 2006 [8,9,10]. The results of the published studies investigating the duration of the operation for classical and laparoscopic surgery are inconsistent. Some of the authors emphasize the superiority of minimally invasive methods in the case of bilateral inguinal hernia repair [11,12,13]. The purpose of our study was to compare the duration of the unilateral inguinal hernia repair depending on the method used.

## 2. Materials and Methods

### 2.1. Patients and Methods

We performed a retrospective review of medical records of all patients who underwent inguinal hernia repair at the Pediatric Surgery, Urology and Traumatology Department of the Regional Hospital in Kielce between January 2011 and December 2018. The study was approved by the institutional research committee (Jan Kochanowski University Research Ethics Board, nr 12/2016) and was performed according to the Declaration of Helsinki’s guidelines.

Patients with bilateral inguinal hernias and those who underwent conversion from laparoscopic to open method or had additional problems (e.g., cryptorchidism, phimosis, hydrocele) were excluded from the study population. There were 19 patients excluded from the study because they required conversion from the PIRS method to an open approach (OA) method. The main reasons for converting from a PIRS to an OA were difficulties in identifying the anatomical structures (because of the presence of a large hematoma at the internal inguinal ring obstructing the view), a large size hernia in very young boys hampering the ability to tightly close the canal, insufflation of gas into the scrotum through an open canal, and a significant tissue edema secondary to an incarcerated hernia. In all those cases, the operating surgeon’s judgement was that the PIRS method was insufficient to treat the patient. All surgeries were performed by experienced surgeons or by a surgical fellow under the close supervision of an experienced surgeon. There were no written criteria for assigning patients to either the OA or PIRS method groups. Choosing the method of surgery was left at the discretion of the surgeon assigned to the case.

The patients’ clinical characteristics, such as the location of the hernia (left- or right-sided), the duration of the surgery, and operative method, were collected from the medical records. 

### 2.2. Operative Techniques 

Open approach inguinal hernia repair was performed using a modified Girard method. After cutting the skin, the subcutaneous tissue, and the aponeurosis of the external oblique muscle parallel to the inguinal ligament, then the spermatic cord (in males) and hernial sac were dissected free and opened. The contents were then transferred to the abdominal cavity, and the opening was closed with absorbable 4-0 sutures. The spermatic cord was relocated below the layer of stitches holding the internal oblique muscle and the transverse fascia with the inguinal ligament. Next, the inguinal ligament was strengthened with two layers of braided, absorbable 2-0 and 3-0 sutures, and the subcutaneous tissue was stitched with a single, braided, absorbable 4-0 suture. Lastly, the wound was closed with continuous intradermal sutures using braided, absorbable 4-0 or 5-0 sutures.

The minimally invasive approach was performed using the PIRS technique. The procedure required one 3.5 mm or 5 mm optical port to be inserted through the umbilicus (in some cases, an additional port was used due to technical difficulties). The peritoneal cavity and inguinal canal openings were assessed using a 30-degree camera. The internal inguinal ring’s opening was closed using braided, nonabsorbable 2-0 sutures, and the procedure was completed with an umbilical wound closure. During the surgery, the intraabdominal pressure was maintained at 8 to 12 mmHg, depending on the patient’s age. The entire operation was recorded. The surgery duration was defined as the time interval in minutes from surgical skin preparation to wound dressing.

### 2.3. Statistical Analysis

The operative was analyzed with respect to the method used and grouping variables: gender and hernia location. The statistical analysis was performed using the Statistical Package Statistica software (version for Windows 13.1 TIBCO Software Inc.—StatSoft, Poland). The study results were presented in the form of the distribution, frequency, medians, and the interquartile range of the studied variable. The normality of the distribution of the analyzed data was checked using The Shapiro–Wilk test. Differences between the studied groups were verified using the nonparametric U Mann–Whitney test. The significance level was set at α = 0.05. 

## 3. Results

A total of 878 patients ages 0 to 18 were included in the study. There were 189 females and 689 males. The patients were divided into two groups by the method of the surgical technique applied. The first group consisted of 701 (79.84%) patients operated with an open approach (OA). The second group was composed of 177 (20.16%) patients operated with the PIRS procedure. Mean age at the time of surgery in the laparoscopic approach (PIRS) group was 5.47 years (range 0–18) for females and 5.36 years (range 0–18) for males. For the OA group the mean age at the time of surgery was 6.34 years (range 0–17) and 4.60 years (range 0–18) for females and males, respectively. 

Analysis of the collected data for studied variables showed that there were statistically significant differences between groups. The duration of the operation for the OA method was longer in comparison to the PIRS method. The difference in medians reached 10 min and was statistically significant (*p* = 0.000). The location of the hernia had a significant effect on the duration of the operation. It took longer to complete the surgery if the hernia was left-sided (*p* = 0.024) (Table 1, Figure 1).

When analyzed by gender, the data showed that the operation duration was statistically significantly longer in boys for both surgery methods. The medians’ difference reached 5 min with *p* = 0.005 and *p* = 0.000 for the OA and the PIRS methods, respectively (Table 2, Figure 2).

In the group of females, the operation duration was 10 min shorter for the PIRS method, and the difference was statistically significant (*p* = 0.000). The hernia location did not affect the duration of the operation if all surgeries were analyzed together (*p* = 0.260) (Table 3, Figure 3).

Location of the hernia did not have an effect on the duration of the operation in females for either method of the surgery with *p* = 0.496 for the OA method and *p* = 0.538 for the PIRS method, respectively (Table 4, Figure 4).

In the group of male subjects, duration of the inguinal hernia repair performed by the PIRS method was 10 min shorter than with the OA method (*p* = 0.000). Analysis of the hernia location as a variable showed no significant difference in terms of the time needed to complete the surgery between left- and right-sided hernias (*p* = 0.085) (Table 5, Figure 5). 

For the OA method, the inguinal hernia repair operation duration in males was not statistically significantly different between left- and right-sided hernias (*p* = 0.428). However, if a male subject’s surgery was performed using the PIRS method, the operation duration was significantly longer for a left-sided inguinal hernia, and the difference was statistically different (*p* = 0.033) (Table 6, Figure 6).

Mean follow-up time was 62.3 months (range 25–98) for females and 65.2 months (range 24–113) for males in the OA group. In the PIRS group, mean follow-up time for females was 64.1 months (range 26–99) and for males 68.5 months (range 25–100).

There were four cases (2.26%) of recurrence in the PIRS group and nine cases of recurrence (1.28%) in the OA group. No cases of testicular atrophy were reported in either group. Because of wound infection at the incision site, 21 patients in the OA group were treated with antibiotics. Two cases of post operation hydroceles and nine cases of hematoma at the internal inguinal ring were reported in the PIRS group but did not require additional intervention. 

## 4. Discussion

Published research studies investigating the inguinal hernia repair duration in children based on the method used show inconsistent results. Some studies report a shorter duration for the minimally invasive laparoscopic approach (LA), others report no statistical difference in the operative time, and there are also studies demonstrating a shorter duration of the procedure using the LA in the case of bilateral hernias.

The results of the prospective study of Kara et al. were similar to ours. The comparison of the length of the operative time for the LA and the OA showed that the duration of the operation was statistically significantly shorter for the LA [14]. Zhang et al. published retrospective study results comparing the two methods of hernia repair in pediatric patients from two medical centers in China and concluded that the LA approach allows completing the surgery faster [15]. The study results by Bertozzi et al. comparing their contemporary experience with the LA to historical controls, when surgery was performed with the OA, suggests that operative time with the LA method is shorter [16]. Thomas et al. reported their experience with the PIRS method and stressed the learning curve’s importance [17].

On the other hand, Shehata et al. demonstrated that hernia repair with the LA might be safely performed even in very young patients, but operative time is comparable to the OA. A LA might be superior to an OA in the case of a clinically suspected or proven bilateral inguinal hernia [18]. 

Similarly, a systematic review and meta-analysis published by Yang et al. revealed that contrary to our findings, a hernia repair’s operative time did not significantly differ between the LA and the OA for unilateral lesions. It was significantly shorter, though, for the LA if the hernia was bilateral [19]. The literature review published by Esposito et al. had similar conclusions. For unilateral hernias, there is no time benefit with a LA versus an OA surgery. In contrast, in patients with bilateral disease, the LA was significantly faster than the OA [3]. Jessula et al., in their summary of the evidence for the laparoscopic pediatric inguinal hernia repair in children, also concluded that operative time for unilateral hernias is no different between the LA and the OA method. However, the LA offers faster operative time if the lesion is bilateral [20]. The discrepancy between operative times for unilateral versus bilateral lesions might be explained by the fact that with the LA method, the same set of ports is being used for repairing both sides, whereas, during an OA surgery, separate incisions are required. It is suggested that the duration of suturing abdominal layers compensates for the initial time of establishing pneumoperitoneum in an LA [21]. Similar observations can be found in our previous work [10].

Interestingly, analysis of our data by gender showed that operative time was statistically significantly longer in males for both methods. For both groups duration of surgery was shorter if the procedure was performed laparoscopically. L. L. Zhu. et al., who also presented the outcomes separately for females and males, had different results. There was no difference between the groups in cases of unilateral hernias for laparoscopic surgery, but operative time was shorter in the group of females if the lesion was bilateral [22]. With an OA, surgery time needed to complete the procedure was shorter in females for unilateral lesions, but there was no statistically significant difference if the hernia was bilateral. In females with a unilateral hernia, operative time was not statistically different between an LA and an OA. However, if the lesion was bilateral, the LA method was quicker. Contrary to our findings, for the group of males in cases of unilateral hernias, operative time was shorter with an OA surgery.

Peng has made a fascinating observation, similar to ours, that the duration of left-sided hernia surgeries is longer than right-sided ones using the laparoscopic method [23]. It might be explained by the fact that left inguinal repair is more difficult for a right-handed person standing on the right-side of the patient. All surgeons were right-handed in our material. 

This analysis should be interpreted in light of its limitations. Due to this study’s retrospective nature, statistically significant differences can best be defined as associations of the outcome rather than a prediction. Furthermore, this study was not randomized, so there is a significant selection bias. Operative method selection was based on the surgeon’s operative method preferences. Finally, due to local practice, the operative time was rounded up to the nearest 5-min interval.

## 5. Conclusions

The operative time for a unilateral inguinal hernia repair using the video assisted and minimally invasive PIRS method is shorter in comparison to the classical open approach method of surgery for both genders. Operative time in females is shorter in comparison to males, and this is true for both methods. Time of surgery for the PIRS method of a left-sided inguinal hernia is longer than surgery of a right-sided inguinal hernia. Therefore, the take home message from our study is that the PIRS method of inguinal hernia repair in children is faster than the OA method and it might be considered a preferred way to perform surgery, especially in females. 

## Figures and Tables

**Figure 1 jcm-10-01293-f001:**
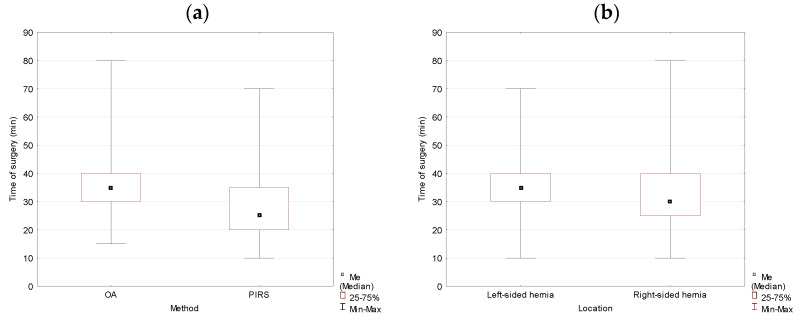
The duration of the operation analyzed by method (OA vs. PIRS) (**a**) and location (left- vs. right-sided inguinal hernia) (**b**) in the studied population. OA, open approach; PIRS, Percutaneous Internal Ring Suturing.

**Figure 2 jcm-10-01293-f002:**
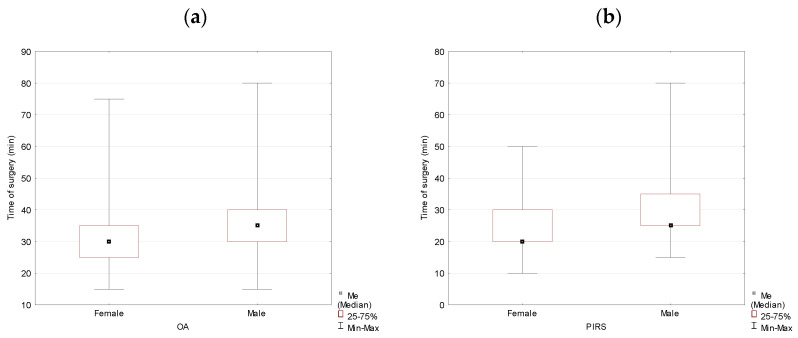
Duration of the operation for OA (**a**) and PIRS (**b**) surgery analyzed by gender (males vs. females) in the studied population.

**Figure 3 jcm-10-01293-f003:**
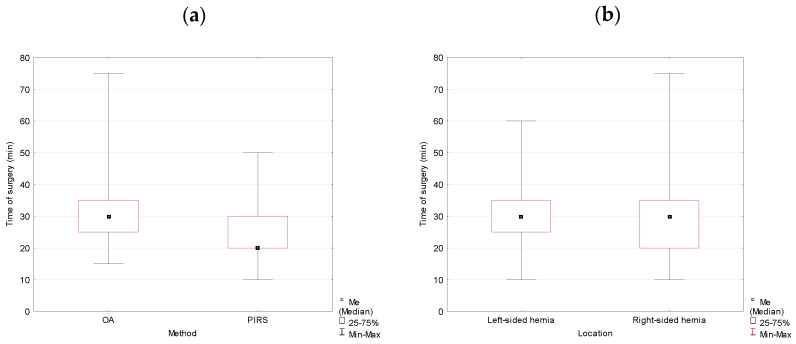
Duration of the operation analyzed by the method of surgery (OA vs. PIRS) (**a**) and location (left- vs. right-sided inguinal hernia) (**b**) in females.

**Figure 4 jcm-10-01293-f004:**
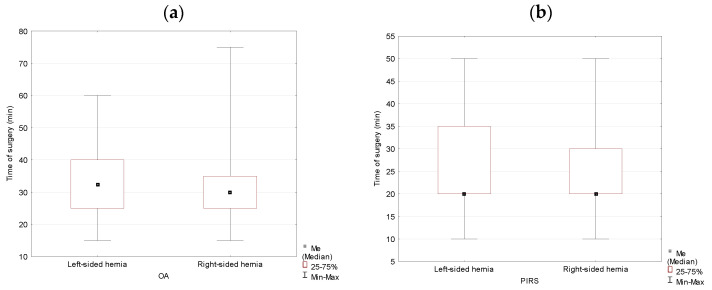
Duration of the operation for OA (**a**) and PIRS (**b**) surgery analyzed by location (left- vs. right-sided inguinal hernia) in females.

**Figure 5 jcm-10-01293-f005:**
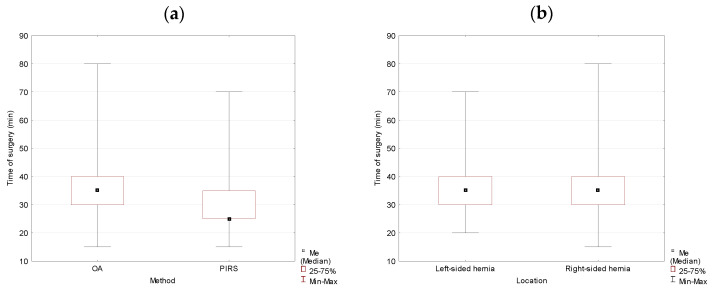
Duration of the operation analyzed by the method of surgery (OA vs. PIRS) (**a**) and location (left- vs. right-sided inguinal hernia) (**b**) in males.

**Figure 6 jcm-10-01293-f006:**
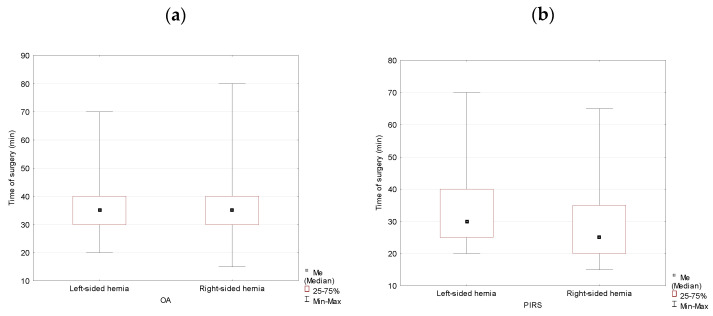
Duration of the operation for OA (**a**) and PIRS (**b**) surgery analyzed by location (left- vs. right-sided inguinal hernia) in males.

**Table 1 jcm-10-01293-t001:** Comparison of the duration of the operation for open approach (OA) vs. Percutaneous Internal Ring Suturing (PIRS) surgery, and for left- vs. right-sided inguinal hernia performed in the studied population.

Variable	*n* (%)	M (SD) Minutes	Me (IQR) Minutes	*p*-Value
Total	878 (100.00)	33.70 (10.52)	30 (15)	
OA	701 (79.84)	35.13 (09.75)	35 (10)	0.000 *
PIRS	177 (20.16)	28.02 (11.51)	25 (15)	
Left-sided hernia	335 (38.15)	34.57 (10.07)	35 (10)	0.024 *
Right-sided hernia	543 (61.85)	33.17 (10.76)	30 (15)	

* U Mann–Whitney test, *p* < 0.05 (α = 0.05). M (SD), mean (standard deviation); Me (IQR), median (interquartile range).

**Table 2 jcm-10-01293-t002:** Duration of the operation for OA and PIRS surgery analyzed by gender, males vs. females, in the studied population.

Variable	*n* (%)	M(SD) Minutes	Me(IQR) Minutes	*p*-Value
OA	701 (79.84)	35.13 (09.75)	35 (10)	
Female	115(16.41)	32.96 (9.99)	30 (10)	0.005 *
Male	586 (83.59)	35.56 (9.65)	35 (10)	
PIRS	177 (20.16)	28.02 (11.51)	25 (15)	
Female	74 (41.81)	24.53 (9.66)	20 (10)	0.000 *
Male	103 (58.19)	30.53 (12.10)	25 (10)	

* U Mann-Whitney test, *p* < 0.05 (α = 0.05). M (SD), mean (standard deviation); Me (IQR), median (interquartile range).

**Table 3 jcm-10-01293-t003:** Duration of the operation analyzed by the method of surgery (OA vs. PIRS) and location (left- vs. right-sided inguinal hernia) in females.

Variable	*n* (%)	M(SD) Minutes	Me(IQR) Minutes	*p*-Value
Total	189 (100.00)	29.66 (10.67)	30 (15)	
Method				
OA	115 (60.85)	32.96 (9.99)	30 (10)	0.000 *
PIRS	74 (39.15)	24.53 (9.66)	20 (10)	
Location				
Left-sided hernia	65 (34.39)	30.69 (10.38)	30 (10)	0.260 ^nss^
Right-sided hernia	124 (65.61)	29.11 (10.82)	30 (15)	

* U Mann–Whitney test, *p* < 0.05 (α = 0.05); not statistically significant (^nss^) U Mann–Whitney test, *p* > 0.05 (α = 0.05). M (SD), mean (standard deviation); Me (IQR), median (interquartile range).

**Table 4 jcm-10-01293-t004:** Duration of the operation for OA and PIRS surgery analyzed by the location (left- vs. right-sided inguinal hernia) in females.

Variable	*n* (%)	M (SD) Minutes	Me (IQR) Minutes	*p*-Value
OA				
Left-sided hernia	42 (22.22)	33.57 (9.52)	32.5 (15)	0.496 ^nss^
Right-sided hernia	73 (38.62)	24.53 (9.66)	30 (10)	
PIRS				
Left-sided hernia	23 (12.17)	25.43 (9.99)	20 (15)	0.538 ^nss^
Right-sided hernia	51 (26.98)	24.12 (9.58)	20 (10)	

not statistically significant (^nss^) U Mann–Whitney test, *p* > 0.05 (α = 0.05). M (SD), mean (standard deviation); Me (IQR), median (interquartile range).

**Table 5 jcm-10-01293-t005:** Duration of the operation for OA vs. PIRS surgery and left- vs. right-sided inguinal hernia performed in males in the studied population.

Variable	*n* (%)	M (SD) Minutes	Me (IQR) Minutes	*p*-Value
Total Male	689 (100)	34.81 (10.21)	35 (10)	
Method				
OA	586 (85.05)	35.56 (9.65)	35 (10)	0.000 *
PIRS	103 (14.95)	30.53 (12.10)	25 (10)	
Location				
Left-sided hernia	270 (39.19)	35.50 (9.78)	35 (10)	0.085 ^nss^
Right-sided hernia	419 (60.81)	34.37 (10.46)	35 (10)	

* U Mann–Whitney test, *p* < 0.05, not statistically significant (^nss^) U Mann–Whitney test, *p* > 0.05 (α = 0.05). M (SD), mean (standard deviation); Me (IQR), median (interquartile range).

**Table 6 jcm-10-01293-t006:** Duration of the operation for OA and PIRS surgery analyzed by location (left- vs. right-sided inguinal hernia) in males.

Variable	*n* (%)	M (SD) Minutes	Me (IQR) Minutes	*p*-Value
OA				
Left-sided hernia	236 (34.25)	35.78 (9.27)	35 (10)	0.428 ^nss^
Right-sided hernia	350 (50.80)	35.41 (9.91)	35 (10)	
PIRS				
Left-sided hernia	34 (4.93)	33.53 (12.82)	30 (15)	0.033 *
Right-sided hernia	69 (10.01)	29.05 (11.54)	25 (15)	

* U Mann–Whitney test, *p* < 0.05, not statistically significant (^nss^) U Mann–Whitney test, *p* > 0.05 (α = 0.05). M (SD), mean (standard deviation); Me (IQR), median (interquartile range).

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
