# Peer review of "The Operative Time for Unilateral Inguinal Hernia Repair in Children Performed with Percutaneous Internal Ring Suturing (PIRS) or Open Approach Method"

_jcm, 2021, doi:10.3390/jcm10061293_

Round 1

Reviewer 1 Report

This is a non-randomized and retrospective study on the operation duration for unilateral inguinal hernia repair in children performed with open or laparoscopic approach. The surgical approach was chosen by the surgeon. No follow-up data have been included. Results of the study confirm the data, already reported in the literature, of a shorter duration of laparoscopic procedures, especially performed in females and in the right side.  

Author Response

We would like to thank the reviewer for the comments and suggestions. Follow – up data was included in the results section of the manuscript – line 176-183.

Reviewer 2 Report

In this retrospective study on inguinal hernia repair in children, the operative duration of two different surgical approaches (laparoscopic and open) was compared. The results showed that the duration of the open procedure was 10 min longer in average. The clinical consequences and operative costs (laparoscopic approach must have been more expensive) have not been evaluated or addressed. Important information on complications, follow-up and recurrences are mandatory in the evaluation of the different surgical approaches.

Unfortunately, the authors did not perform an intention to treat analyses. Cases where the laparoscopic approach had to be converted to open procedure was excluded which is a major concern. At least the number of excluded patients should be reported. Surgeon qualification is also missing. Were all patients operated by specialists in pediatric surgery and what were the criteria for choosing the laparoscopic approach? Another problem is that only a minor part of the population was operated with this method (patient selection?).

Another problem is the described operative procedure for the open method. Duplication and “strengthening” of the ileal ligament and the describe closure of the hernia opening seems to differ from general routine with an annular closure in males, only. Both procedures are time-consuming. The same applies for the use of subcutaneous suturing.

To my opinion the only thing that can be concluded is that there was an approximately 10 min difference in operative duration in favor of the laparoscopic approach but more importantly the clinical and economic consequences remain unknown.

Author Response

  1. As requested by the reviewer we have added the data on the group of patients excluded from the analysis on the basis of needing the conversion from PIRS to an open approach method – line 53-54.
  2. In the study population the surgeon assinged to the case was the one making a decision which method of surgery should be used. The reason why only minor part of the population was operated with LA method was because only one member of pediatric surgical team was very experienced with this method and he was the only one performing inguinal hernia repair with LA approach. Information on surgeons’ qualifications and the criteria for choosing laparoscopic approach were added to the manusript – line 55-57.
  3. Authors do understand the reviewer’s concern that adding additional steps to a procedure of closing the inguinal hernia with an open approach might prolong the operation time. However, the technique of open approach method described in the manuscript is a standard operating procedure used by the majority of pediatric surgeons in Poland and approved by the Polish Society of Pediatric Surgeons.

Reviewer 3 Report

This is an interesting paper but i have some criticisms:

  • first, title should be re-written! PIRS is not a lapascoscopic technique bue a video assisted technique. SO you have to add PIRS into the title.
  • the main bias is the age range at operation. there is not comparison nor data about age at surgery. it is not clear why open approach in female should be longer than video assisted technique. 
  • data about patients should be add only into the results section
  • there is not follow-up. How many complications or relapses or recurrent hernia for both group? 
  • the field is interesting , so i suggest to re-write the manuscript focusing on important data such as number of surgeons, age range at surgery, complications and recurrences.

Author Response

  1. As suggested the title was changed to: The Duration of the Operation for the Laparoscopic, Percutaneous Internal Ring Suturing (PIRS) method vs Open Approach for Unilateral Inguinal Hernia Repair in Children
  2. The data on the age of the study subjects at the time of the surgery for both study groups was added in the results section of the manusript– line 99-102.
  3. As for the question why an open approach in female took longer than video assisted technique it is an authors’ opinion that in the hands of an experienced operator laparoscopic method of hernia repair is simply faster in general.
  4. Patients’ data was as suggested moved to the results section – line 95-98.
  5. Follow – up information was added into the results section – line 176-183.

Round 2

Reviewer 2 Report

Happy with the response to my questions and the revison of the manuscriupt

Author Response

We would like to thank the reviewer for the comments and suggestions.

Reviewer 3 Report

the title is not correct. i suggest to change it. the 'duration' is not scientific

again , why did you perform open approach and not real laparoscopic approach in those 19 patients ? this is important ad should be stressed.

your study is about PIRS, so the term laparoscopic should not be cited. video assisted or PIRS should be used during the text. 

the main outcome of your study is to report your experience with PIRS and not with laparoscopic inguinal hernia repair! 

surgeon choice is not scientific! this means that in obese patients you have to use it or not? in big hernia do you use PIRS or open technique? 

another question, do you perform standard laparoscopic inguinal hernia? 

personal experience: i usually perform laparoscopic inguinal hernia ,and i think that PIRS is a good technique in female but very very dangerous in male.

Author Response

  1. As suggested by the reviewer the title was changed to: The Operative Time for Inguinal Hernia Repair in Children performed with Percutaneous Ring Suturing (PIRS) or open approach method.
  2. The reasons for converting from PIRS to open approach were described in the manuscript – line 54 to 60.i
  3. As suggested instead of the laparoscopic we used PIRS as a description of the procedure.
  4. As You have poined out outcome of the study should be reported as an experience with PIRS not laparoscopic repair. We’ve corrected that in the manuscript.
  5. We agree that surgeon’s choice is not a scientific way to select a patients for the study. It is certainly a major limitation of our study and reviewer is correct that from our data recommendations as to which procedure to use in a specific clinical situation (as You have mentioned obesity or large size of hernia) cannot be made. However it is also true that obesity, age of the patient or size of the hernia were not an exclusion criteria. Those clinical characteristics were comparable in both groups.
  6. Since 2018 in Our institution classical laparoscopic inguinal hernia repair was performed in boys 16 years old and older. Those patients were not included in the study.
  7. In our opinion PIRS might be considered as a gold standard in females and is indeed more difficult in boys. In our experience the most difficult part is visualization of the spermatic cord which is sometimes impossible to do even when additional port is utilized. In those cases we simply convert the surgery to open approach.